# Cross-Cultural Adaptation and Reliability of the Back Pain and Body Posture Evaluation Instrument (BackPEI) to the Spanish Adolescent Population

**DOI:** 10.3390/ijerph18030854

**Published:** 2021-01-20

**Authors:** Vicente Miñana-Signes, Manuel Monfort-Pañego, Joan Morant, Matias Noll

**Affiliations:** 1Body Languages Didactics Department, Academic Unit of Physical Education, Teacher Training Faculty, University of Valencia, 46022 Valencia, Spain; manuel.monfort@uv.es (M.M.-P.); joanmorant1994@gmail.com (J.M.); 2Physical Education Department, Instituto Federal Goiano, Rialma, Goias 76310-000, Brazil; matias.noll@ifgoiano.edu.br

**Keywords:** questionnaire, back health, assessment, cross-cultural adaptation, adolescents, secondary school

## Abstract

The prevalence of back pain (BP) among children and adolescents has increased over recent years. Some authors advocate promoting back-health education in the school setting. It is therefore important to adopt a uniform suite of assessment instruments to measure the various constructs. The present study aimed to perform a cultural adaptation of a validated measurement instrument (BackPEI), beginning with a translation and cultural adaptation phase, followed by a second phase to test reliability using a test-retest design. The translation and cross-cultural adaptation were performed based on the guidelines. Reliability was tested by applying the questionnaire to 224 secondary school students, at two different times with a 7-day interval between the tests. In general, the Spanish version presented adequate agreement for questions 1–20, with only question 9 achieving a low Kappa range of 0.312 (−0.152–0.189). The question about pain intensity did not show differences between the test means (4.72 ± 2.33) and re-test (4.58 ± 2.37) (*p* = 0.333), and the responses for these two tests obtained a high correlation (ICC = 0.951 (0.928–0.966); *p* = 0.0001). Psychometric testing indicated that the Spanish version of the BackPEI is well-adapted and reliable, based on the test–retest design, providing similar results to the original Brazilian version.

## 1. Introduction

The prevalence of back pain (BP) among children and adolescents has increased over recent years [1,2]. BP is a symptom, such as headaches and dizziness [3], rather than a disease, and it is associated with a wide range of risk factors [4,5], among which we highlight the sedentary positions caused by prolonged sitting on school days, as well as at home in front of computers or other devices, and the inappropriate postural habits of students in their daily lives [6,7]. Moreover, it is known that from the ages of 10 and 14, these episodes of discomfort begin to be experienced in a significant way [8,9,10], which may have repercussions in adulthood [11].

Some authors and organizations advocate promoting the back health education research line and especially its implementation in school-based education programs [12,13,14]. However, in order to reach robust conclusions concerning posture interventions in schools, it is important to adopt a uniform suite of assessment instruments to measure the various contents in the investigations (BP prevalence, general and specific back care knowledge, daily postural habits, the student’s and teachers’ perceptions, etc.) [15].

To the best of our knowledge, there are only two validated and reliable self-reported evaluation instruments concerning postural habits within an educational orientation and content: The Back Pain and Body Posture Evaluation Instrument (the BackPEI questionnaire) [16], and the Back-Health Related Postural Habits in Daily Activities (the BEHALVES questionnaire) [17].

The BackPEI questionnaire [16] was designed to identify the presence of BP in the last 3 months before its application. It studies questions on the occurrence, frequency, and intensity of the back pain. The intensity of the pain, question number 21, was assessed using the visual analog scale (VAS), it is a 10-cm horizontal line in which “0” means ‘‘No Pain’’ and “10” means ‘‘The Worst Pain I Can Imagine’’. The first 20 questions are closed, which is only possible on response: Questions 1–8 are based on lifestyle; questions 9–14 deal with posture adopted during daily-life activities; questions 15 and 16 ask about parent studies; questions 17–19 are designed to identify the presence of back pain in the last three months, and the occurrence and frequency of the pain.

BEHALVES [17] was developed to assess back health-related postural habits in the daily activities of adolescents. The items were grouped into five categories: Standing posture (items 1–4), sitting posture (items 5–13), use of backpacks (items 14–20), mobilizing heavy weights (items 21–26), and lying posture (items 27–31). The items in the questionnaire were scored with: 1 = Never, 2 = Hardly ever, 3 = Almost always, and 4 = Always.

Although both instruments were developed to study body postural habits in adolescent populations, BERHALVES was validated on Spanish adolescents and BackPEI on Brazilian adolescents. The cultural differences under which both questionnaires were validated means that they are useful only in those setting, so it not possible to compare results from them to apply a criterion validity study. Hence, the cross-cultural adaptation of one of these questionnaires could be the first step to consolidate valid instruments. Only then could a criterion validity study be carried out when comparing one instrument to the other.

Therefore, the present study aimed to perform a cross-cultural adaptation of a validated measurement instrument (BackPEI), beginning with a translation and cultural adaptation phase, followed by a second phase to test the reliability by the test–retest design. In this way, we intend to answer the research question concerning the possibility of preparing a validated assessment tool to contribute to the improvement of back health. It was hypothesized that the Spanish version of the BackPEI presents a degree of agreement (a psychometric test) similar to that of the original validated instrument.

## 2. Materials and Methods

### 2.1. Study Design

A cross-cultural adaptation and a repeatability study were performed.

### 2.2. Participants

#### 2.2.1. Translation

Seven researchers participated in the translation. Three of them carried out the translation: (1) A bilingual, bicultural native Brazilian researcher from the Teacher Training Faculty of the University of Valencia (T1); (2) a bilingual, bicultural, and native Brazilian researcher from the Polytechnic University of Valencia (T2), and (3) a researcher and professor of the University of Valencia (T3). Four of the seven researchers were the authors of the present study who participated as organizers and mediators of the translation process (OMTP).

#### 2.2.2. Cultural Adaptation

An expert committee (EC) made up of two bilingual professors from the “Instituto Federal Goiano”, Brazil, who specialize in physical education teaching, physiotherapy, and public health participated in the revisions of the questionnaires.

#### 2.2.3. Reliability

A sample of 400 students from a public secondary school, chosen based on a convenience factor, from the Valencian Community (Spain) agreed to participate in the study. 176 participants were excluded because: (1) They did not complete the second questionnaire; (2) they had a high number of omitted answers (more than 20% of the questions); and (3) because they forgot to put the control name (for matching purposes) in the written questionnaire. The final sample was 224 students (56% recruited; 15.1 ± 1.4 years old; 48.7% girls, *n* = 109) who participated in this investigation (Table 1). Sousa and Rojjanasrirat [18] recommend at least 10 subjects per item of the instrument. As BackPEI contained 21 questions, then at least 210 participants would be required.

### 2.3. Procedure

#### 2.3.1. Translation Procedure

The translation and cross-cultural adaptation of the original Brazilian version of the BackPEI into a Spanish version was conducted according to the established guidelines [18,19], and it consisted of three steps. Each step of the Delphi-process was reviewed by the OMTP. The first five steps aimed to achieve equivalence between the instrument in the source language (SL; original language of the instrument, Portuguese) and the instrument in the target language (TL; desired language, Spanish).

Step one: The translation (from SL to TL) was performed by T1.

Step two: A blind backward translation (from TL to SL) was carried out by T2, who did not have knowledge of the original Brazilian text, and who produced a translation of the consensus target language version. The purpose of this step was to highlight discrepancies between the source document and the translation.

Step three: T3 compared the target questionnaire with the original version (from SL to TL). The objective in this step was to detect any possible differences between the backward translation and the original version, and to improve these through consensus among the translators.

#### 2.3.2. Cultural Adaptation Procedure

Step four: Then, the EC reviewed the Spanish version (TL) compared to the original (SL). The objective of the committee was the production of a pre-final version for field testing, based on the version obtained from the forward and backward translations. The Delphi method was used. The anonymous responses were aggregated and shared with the group after each round. The experts were allowed to adjust their answers in subsequent rounds. Each committee member compared the Spanish version and original versions of the BackPEI on an item-by-item basis and in general (all items), by scoring the equivalence between the two versions through four questions in terms of semantics (i.e., equivalence in the meaning of words), idiomatic (i.e., equivalence in idioms and colloquialisms), experiential (i.e., equivalence in the target cultural context), and conceptual (i.e., equivalence of the concept and the experiences of the target culture). The questions answered by the experts during this content validity procedure are presented in Table 2.

The equivalence was scored using the surveys to collect the experts’ responses on 5-point Likert scales and by making observations as open-ended questions. The five Likert-scale categories were: (1) Strongly Disagree; (2) Disagree; (3) Neither Agree nor Disagree; (4) Agree; (5) Strongly Agree [20]. The experts expressed their opinion by selecting only one category per statement. After the evaluation of each individual item, all the items evaluated with modifications were discussed and revised to produce the final Spanish version of the BackPEI. Since multiple rounds of questions were asked, and the panel was told what the group thought as a whole, the Delphi method sought to reach the correct response through consensus. This Delphi process needed three rounds.

Step five: Proofreading errors were corrected before the pre-final Spanish version of the BackPEI was produced.

#### 2.3.3. Reliability Test Procedure

Step six: Psychometric testing of the pre-final version of the translated instrument in a sample of the target population. To analyze its reliability, the Spanish pre-final version of the BackPEI was given to 224 adolescents twice (test-retest), with a seven-day interval between each test.

This study was carried out between July 2019 and February 2020. Data collection was undertaken during the second term of 2020 at the school facilities under the supervision of one member of our research group in the presence of Physical Education teachers before the Severe Acute Respiratory Syndrome Coronavirus 2 (SARS-CoV-2) pandemic. The written questionnaires to evaluate the test–retest reliability were administered twice, with a seven-day interval between each test [21], during physical education classes. The student took an average of between 10 and 20 min to complete the questionnaire according to their age.

### 2.4. Instrument

The tool used was the cross-culturally adapted and reliability tested BackPEI questionnaire (16), described in the introduction. Because girls and boys can use some different postures [22], the questionnaire was developed to differentiate the boys’ and girls’ versions.

### 2.5. Ethical Statements

We obtained institutional ethical approval from the Ethics Committee in experimental research from the University of Valencia (reference number: H1529993833413). The students and their parents provided their consent to participate in the study. The school principal also provided written informed consent.

### 2.6. Data Analysis

In the Delphi method, consensus on cross-cultural validating is achieved when at least 80% of the expert panel score an item equal to four or higher (4 = Agree, 5 = Strongly agree). The expert’s responses to the open-ended questions were analyzed to identify, modify, and include: (1) Semantic, idiomatic, experiential, and conceptual equivalences, and (2) to consider any comment related to the study between the two versions (the boys’ and girls’ questionnaires).

A descriptive study was used in the study, as proposed in previous studies [16]. The data from test and re-test procedures for questions 1–20 were analyzed using Cohen’s kappa coefficient (k) for nominal scales with a 95% Confidence Interval, as well as percentages of absolute agreement. Cohen’s kappas could not be calculated when the 2 × 2 tables were not completely filled in (e.g., item number 13). The results were classified as poor (k < 0.2), fair (0.2\k to 0.4), moderate (0.4\k to 0.6), good (0.6\k to 0.8), or very good (k > 0.8) [23]. To include a question in the BackPEI, it needed to obtain a minimum value of k = 0.5 [24]. Arbitrarily, a percentage of absolute agreement of 75% or more was also considered as acceptable reliability. Agreement between the test and re-test for question 21 (pain intensity) was measured in terms of the relationship between the answers, as revealed using the intraclass correlation coefficients (ICC) for repeated measurements with a 95% Confidence Interval. It applied a T-retest intra-rater reliability two-way mixed effects based on mean of multiple measurement with absolute agreement [25]. An ICC of 0.75 or more was considered a measure for acceptable reliability [26]. In order to compare the measures of central tendency obtained in the evaluations the Wilcoxon test was used. The level of significance adopted was 0.05. Data analysis was performed using the SPSS^®^ IBM^®^ software, r. 26 (IBM, Armonk, NY, USA).

## 3. Results

### 3.1. Translation

During the step one (the forward translation) analysis, the OMTP decided not make any changes. Step 2 (backward translation) was identical to the original questionnaire in Portuguese (SL). In the third step, no major problems were encountered during the forward translation and backward translation phases of the Brazilian version, and the OMTP did not suggest amendments to any words or phrases.

### 3.2. Cultural Adaptation

Step 4 (Expert Committee): The committee took three rounds to accomplish the objective of the study, the production of a pre-final version for field testing. Each expert scored 88 items in the respective rounds.

In the first one, all items in the general and item-by-item basis achieved 80% consensus from the expert panel. Only, items 4 and 5 of the Spanish version of BackPEI received a score of (3) Neither Agree nor Disagree concerning the first question for the experts “Is the Spanish translation of the Portuguese version adequate? Observations”. One of the experts stated that: “The construction is confusing in both questions. Does “sits down” (se sienta, in Spanish) means how many hours the person watches TV while seated?”. In addition, the expert panel suggested changing the use of “usted” for the second person pronoun used idiomatically for closeness or familiarity (3rd question in the experts’ questionnaire, Table 1).

The second-round survey included changing the term “sit down” to “remain” (permanences, in Spanish) and using the second person pronoun in all the items. All the items in the general and item-by-item basis achieved 80% without exceptions. However, the expert panel suggested more modifications to the items. It was necessary to change the female gender use in the questionnaire for girls in items 4 and 5 (permaner sentada, in Spanish). Modifications were also suggested for: Item 14 “¿cómo sueles llevar la mochila?” (in Spanish) to keep the pattern of the other questions; items 15 and 16 “¿Cuál es el grado de formación de tu padre/madre/tutora?” (in Spanish); and item 19 includes the Spanish pronoun “te ocurre”.

In the last round, all the suggestions made in the previous round were incorporated. The expert panel achieved 100% consensus, and scored all the items equal to five, the highest level (5 = Strongly agree) (Table 3).

Step 5 (proofreading): No grammatical or spelling errors were found. In the sixth step, the pre-final version for the pre-test was ready.

### 3.3. Psychometric Test

According to the kappa coefficient for questions 1–20 in the questionnaire, 5 were classified as ‘‘very good’’, 8 as ‘‘good’’, 1 as “moderate”, and 1 as “fair” (Table 4). The answer rate was high, with a missing value of less than 10 %. Based on question 21 (*n* = 115), there were no differences in the intensity of pain between test means (4.72 ± 2.33) and re-test (4.58 ± 2.37) based on the Wilcoxon test (*p* = 0.333), and the responses for these two tests were highly correlated (ICC = 0.951, (0.928– 0.966); *p* = 0.0001).

## 4. Discussion

Our study aimed to perform a cultural adaptation of a validated measurement instrument (BackPEI), with an initial translation and cultural adaptation phase, followed by a second phase to test reliability by using a test–retest design. Our main finding indicates that the Spanish version of the BackPEI is reliable and presents similar results to the original Brazilian version, hence being relevant for epidemiological research when comparing countries where the Portuguese and Spanish languages are spoken.

According to the guidelines [18,19], the process of cross-cultural adaptation of the Spanish BackPEI version followed the recommendations proposed, with some adaptations. The cross-cultural adaptation must be the production of several translations by, at least, two independent translators. This leads to the detection of errors and divergent interpretations of ambiguous items in the original tool [19]. Using only a single translator is far from ideal [27]; however, because of the simple and short type of questions in the BackPEI, we only required four independent bilingual translators, one for the forward translation (first step) and another for the back translation (second step), and two for the cultural adaptation. Because we only had one translator in the first step, we did not synthesize the results of the translation before the second step.

Regarding the consensus (Table 4), for ten items the k values for items 1–20 in the questionnaire and percent agreement were good (practice of physical exercise, frequency of physical exercise, competitive or non-competitive physical exercise, preferred sleeping position, position adapted when lifting an object from the floor, mode of transporting the school backpack, mother’s level of education, father’s level of education, parents with a history of back pain, and presence of back pain). For two other tests, percentages of absolute agreement were below 75%, but their kappa values were good (reading and/or studying in bed, and time slept each night) and can therefore be considered reliable as well. For five tests, the kappa’s were below 0.60 and the percentage of absolute agreement above 75% (time spent each day using a computer, sitting position when writing, sitting position on a chair when talking, sitting position when using a computer, and impeding the performance of activities) and they are also therefore considered reliable. Finally, for two tests (time spent watching TV and frequency of back pain), the kappas as well as the percentage of absolute agreement showed moderate values, and they were below the criteria for acceptance.

Specifically, the four questions on the sitting position (item 4, 9, 10, and 11) obtained the lowest kappa values, coinciding with the original version of the BackPEI [16] and the Turkish version [28]. In the target version, the results were less than 0.4 for items 9 and 10, but they achieved a high agreement (86%). In the three versions of the BackPEI, the lowest concordance score was achieved by item 9. This result could be due to the fact that the type of response was multiple choice [29]. Throughout the school day, as well as the day in general, we adopt many correct positions and therefore there could be various responses affecting agreement. Some research suggests that people with LBP assume more static, sustained end-range postures while sitting, and use large infrequent shifts in posture rather than small, subtle spinal movements regularly [30,31]. As a result, dynamic sitting approaches which facilitate subtle spinal motion have been proposed as a means of reducing LBP during sitting [32]. Question 21, concerning the intensity of the pain, did not show differences between the averages for the test and re-test, and similar and high correlated results were found in both tests (ICC > 0.93; *p* = 0.05). Both tests achieved around 4.5 to 4.7 points on the VAS (item 21), described as a moderate pain [33]. In other studies, initially the intensity of back pain in adolescents is usually low [34].

Concerning populations, the two translated versions, the Turkish and the Spanish, were applied to a sample of secondary school students. With regard to the test–retest interval, the original and Spanish versions were administered twice with a seven-day interval between each test, while in the Turkish version they used a two-week interval [28]. One week may be considered sufficient time for students to forget the answers they gave in the previous week; however, this period of time is insufficient for changes in their daily habits [24]. Despite the fact that in the Turkish BackPEI version, the authors [28] used Cohen’s kappa coefficient for question 21 (intensity of back pain), based on the current literature [35], we believe the statistical test used in the present study (Wilcoxon test) is much more appropriate, as this question gives us data from a continuous variable.

To the best of our knowledge, only two questionnaires have been validated based on back postural habits in children and adolescents [16,17]. However, the BackPEI not only evaluates the postural habits adopted by school-age children during activities of daily life (ADLs), but also the possible risk indicator associated with this situation, such as the prevalence of BP. This allows it to be considered as a comprehensive instrument that considers both dependent, independent, and confounding variables using just 21 questions and five categories: (a) Practice physical exercise (*n* = 3); (b) active lifestyle (*n* = 5); (c) postural habits (*n* = 6) of which sitting postures (*n* = 3), lifting weights (*n* = 1) and using backpacks (*n* = 2); (d) parents’ level of education (*n* = 2) and (e) back pain (*n* = 5). Moreover, the BackPEI is distinguished from the others because it was developed with a version for boys and another for girls. Besides, it was translated into English, although the reproducibility of the English version has not yet been assessed. Besides, as the studies suggest [18], to be able to use the BackPEI in English a complete process of translation and cross-cultural adaptation would be needed.

In order to continue improving evaluation tools for assessing back health in the school setting, the validity of an instrument has to be analyzed through content validity, criterion validity, and construct validity [36]. Criterion validity refers to the degree to which the instrument produces results similar to those of other existing and valid instruments/equipment to evaluate the same construct [37]. Thus, criterion validity could be the next measurement test that both the BackPEI and the BEHALVES [17] could pass. Based on age, perhaps the BackPEI could be used as a referrer, and the results between the two tools could be compared.

On the subject of the implications for teachers and school health, it is important to have questionnaires as tools to assess postural habits, as well as risk factors in children and adolescents in the school setting. Moreover, the BackPEI is a quick and effective instrument to screen as students only need 10–20 min to complete it.

### Limitations

Since the panel size was neither representative of any population nor statistically calculated, and the recruiting of the experts was a subjective process, our results should not be interpreted as representing the views of all the experts of the studied fields. The number and specialty of the translators could be improved. Despite having carried out the double translation, there is currently no rigorous evidence of the value of backward translation in questionnaire adaptation, leading to suggestions that it could be omitted [38]. Moreover, pilot testing of the pre-final version of the instrument in the target language with a monolingual sample and another pilot testing with a bilingual sample could be developed. The results of this study should be viewed with caution, as it is a convenience sample.

## 5. Conclusions

Psychometric testing indicated that the Spanish version of the BackPEI is well adapted and has a reliable test–retest design which provided similar results to the original Brazilian version. The Spanish BackPEI version represents an instrument with the same conditions as the Brazilian version for evaluating back pain in adolescents as well as body postural in school.

## Figures and Tables

**Table 1 ijerph-18-00854-t001:** Descriptive data of the sample.

Age (Years)	Gender	*n*	X Weight (kg) (±SD)	X Height (cm) (±SD)
13	F	12	42.7 (±7.2)	1.54 (±0.05)
	M	14	46.9 (±11.9)	1.55 (±0.12)
14	F	28	47.4 (±7.5)	1.62 (±0.07)
	M	31	55.9 (±12.6)	1.66 (±0.08)
15	F	31	52.5 (±8.1)	1.64 (±0.06)
	M	27	60.2 (±9.5)	1.73 (±0.07)
16	F	20	54.2 (±8.9)	1.63 (±0.07)
	M	15	66.6 (±11.7)	1.74 (±0.08)
17	F	17	54.0 (±5.3)	1.62 (±0.08)
	M	23	68.5 (±12.0)	1.76 (±0.07)
18	F	1	57.00	1.70
	M	5	73.0 (±6.7)	1.77 (±0.07)

F: Female; M: Male; X: Mean; SD: Standard deviation; cm: Centimeters.

**Table 2 ijerph-18-00854-t002:** Assessment instrument questions for experts.

Questions for Committee of Experts
1. Is the Spanish translation of the Portuguese version adequate? Observations
2. Is the vocabulary used in the Spanish version of the questionnaire correct and understandable in relation to the Portuguese questionnaire? Observations.
3. Is the style and registration of the questionnaire translated into Spanish faithful to the original Portuguese version? Observations.
4. Do you consider that the questions asked fit into the culture of Portuguese/Brazilian speakers? Observations.

**Table 3 ijerph-18-00854-t003:** Results of the three Spanish versions of the Back Pain and Body Posture Evaluation Instrument (BackPEI) carried out by the Expert Committee using the 5-point Likert scale.

Domains	1st Round		2nd Round		3rd Round	
	X	SD	X	SD	X	SD
Semantic-translation	4.81	0.269	4.83	0.236	5.00	-
Conceptual and understanding	4.93	0.101	5.00	-	5.00	-
Idiomatic	4.50	0.707	5.00	-	5.00	-
Experiential	5.00	-	5.00	-	5.00	-

X = mean; SD = standard deviation.

**Table 4 ijerph-18-00854-t004:** Results of the Cohen’s kappa coefficient for the 20 questions in the Spanish version of BackPEI. Reliability tested by applying the questionnaire at two different times with a week interval.

Q	Title of the Question	*n* (Missing Data)	Reliability 7-day Interval	
			A	*k* Value (IC 95 %)
1	Practice of physical exercise	224 (0.0%)	96.9%	0.918 (−0.031–0.060)
2	Frequency of physical exercise	223 (0.4%)	87.8%	0.834 (−0.034–0.060)
3	Competitive or non-competitive physical exercise	224 (0.0%)	93.3%	0.897 (−0.028–0.052)
4	Time spent watching TV	224 (0.0%)	68.8%	0.522 (−0.069–0.096)
5	Time spent each day using a computer	224 (0.0%)	78.6%	0.583 (−0.096–0.101)
6	Reading and/or studying in bed	224 (0.0%)	67.4%	0.669 (−0.054–0.084)
7	Preferred sleeping position	216 (3.6%)	89.8%	0.827 (−0.039–0.068)
8	Time slept each night	222 (0.9%)	66.7%	0.651 (−0.052–0.080)
9	Sitting position when writing	224 (0.0%)	86.6%	0.312 (−0.152–0.189)
10	Sitting position on a chair when talking	224 (0.0%)	86.2%	0.378 (−0.141–0.182)
11	Sitting position when using a computer	224 (0.0%)	88.4%	0.550 (−0.110–0.159)
12	Position adopted when lifting an object from the floor	224 (0.0%)	92%	0.762 (−0.062–0.105)
13	Carrying school material	224 (0.0%)	100%	1.000 (-)
14	Mode of transporting the school backpack	224 (0.0%)	98.6%	0.762 (−0.135–0.234)
15	Mother’s level of education	223 (0.4%)	78.8%	0.729 (−0.043–0.070)
16	Father’s level of education	224 (0.0%)	79.8%	0.743 (−0.041–0.068)
17	Parents with a history of back pain	224 (0.0%)	79.9%	0.689 (−0.052–0.082)
18	Presence of back pain	224 (0.0%)	88.8%	0.792 (−0.043–0.074)
19	Frequency of back pain ^a^	118 (0.0%)	61.1%	0.528 (−0.077–0.109)
20	Impeding the performance of activities ^a^	107 (9.3%)	76.6%	0.517 (−0.110–0.155)

Q = questions; A = Agreement; *k* = Cohen’s kappa coefficient (k); ^a^ related with them who answered ‘yes’ in the item number 18 (*n* = 118).

## Data Availability

The data presented in this study are available on request from the corresponding author. The data are not publicly available due to ethical restrictions.

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
