# Peer review of "Cross-Cultural Adaptation and Reliability of the Back Pain and Body Posture Evaluation Instrument (BackPEI) to the Spanish Adolescent Population"

_ijerph, 2021, doi:10.3390/ijerph18030854_

Round 1

Reviewer 1 Report

Thanks to the authors for considering the suggestions made in the first review, I believe that this manuscript has been improved and clarified after its review.

Author Response

13/01/2021

Dear Reviewer:

We appreciate your reviews and attention. Your suggestions helped us improve the quality of our paper.

We are very grateful.

Sincerely,

Vicente Miñana-Signes

1 Body Languages Didactics Department. Academic Unit of Physical Education. Teacher Training Faculty. University of Valencia. Av. dels Tarongers, 4, 46022, Valencia. Spain.

*Correspondence to: V. Miñana. E-mail: vicente.minana@uv.es

Reviewer 2 Report

Thank you for providing a revised version of your manuscript. All but one of my previous comments were properly addressed. A minor pending issue regards the ICC reporting.

Statistical analysis. Thank you for informing the ICC results are presented for repeated measurements. However, the ICC[k,q] model was still not reported. There are several models for calculation of ICC (e.g., https://doi.org/10.1371/journal.pone.0219854,http://dx.doi.org/10.1016/j.jcm.2016.02.012) and this should be made explicit.

Author Response

Responses to Reviewers' Comments

[IJERPH] Manuscript ID: ijerph-909944

We thank the reviewers for their thoughtful and in-depth comments concerning our manuscript. Your suggestions helped us improve the quality of our paper. We carefully considered every comment and made the appropriate changes, which are highlighted in red font. Our response is noted below.

Reviewer #2:

Thank you for providing a revised version of your manuscript. All but one of my previous comments were properly addressed. A minor pending issue regards the ICC reporting.

Statistical analysis. Thank you for informing the ICC results are presented for repeated measurements. However, the ICC[k,q] model was still not reported. There are several models for calculation of ICC (e.g., https://doi.org/10.1371/journal.pone.0219854,http://dx.doi.org/10.1016/j.jcm.2016.02.012) and this should be made explicit.

Author’s response: According to the article of Koo and Li (2015) we have applied a T-retest intra-rater reliability two-way mixed effects based on mean of multiple measurement with absolute agreement

Koo, T. K., & Li, M. Y. (2016). A guideline of selecting and reporting intraclass correlation coefficients for reliability research. Journal of chiropractic medicine15(2), 155-163.

This manuscript is a resubmission of an earlier submission. The following is a list of the peer review reports and author responses from that submission.

Round 1

Reviewer 1 Report

Thank you very much for the opportunity to review this document, the authors have done important work about an interesting field related with the pain back in adolescent. Some aspects could be review to add better understanding to reader, so the authors are suggested to review and consider the following points.

TITLE

The title could be more descriptive of the work done. It could confuse its ending "for Spanish". “Cross-cultural adaptation to Spanish…” is suggested. At the end of the title the authors could specify the sample or type of people who have been studied (adolescents, students of secondary ...).

ABSTRACT

It is suggested review of the abstract according to the changes made in the manuscript.

INTRODUCCTION

Line 30. “These symptoms (3)”. Could be interesting to do mention of the symptoms more important and avoid to readers search the reference 3.

Line 32. “…other IT devices”. In general this expression can be interpreted as other informatics devices, but seem better say that instead of IT.

Lines 42 to 44. It is suggested: Back Pain and Body Posture Evaluation Instrument (BackPEI questionnaire), and the Back-Health related postural Habits in Daily Activities (BEHALVES questionnaire).

Lines 46 to 51. In this point is necessary to made to the authors  a call about the concepts of process of validation of an instrument of measure, cross-cultural adaptation, validity and reliability, because in this paragraph and across of the manuscript these concepts appear all mixed and are shown with some confusion, according to knowledge of this reviewer.

METHODS

Study design. The statement is not considered appropriate. Cross-cultural adaptation is independent test of reliability, both require different research processes and cultural adaptation cannot be considered based on a test-retest. Here is shown certain confusion methodologic about the concepts of cross-cultural adaptation and reliability. It is proposed to reflect to the authors that this study aim be a validation study of a measurement instrument, with a first cultural adaptation phase and at least another phase that tested one of the reliability aspects, in this manuscript was made a test-retest (there are some more). We continue talking about these aspects in the following paragraphs.

After seeing the methods, it was found that the validation process carried out in this study was developed in six steps. All these steps cannot be considered as the process of cross-cultural adaptation, from our point of view, are identify three different moments of investigation. Steps 1 to 3 were the Translation and cross-cultural adaptation of the BackPEI from Portuguese to Spanish. The fourth and fiftieth steps seem like a content validity process carried out by an expert group of Brazilian teachers. The sixth step was a psychometric test where reliability was studied by test-retest in a sample of students. If the authors share this three-moment research approach, they should take it into account in their methods, results, and discussion.

In the case that steps 4 and 5 were considered as part of cross-cultural translation and adaptation and not as an independent content validity step, a question arises. The Committee of Experts was made up of Brazilian teachers and taking into account that the instrument will be applied in the Spanish population, why were the semantic, idiomatic, experiential or conceptual aspects not analyzed by Spanish experts? Could be that a limitation? How many experts collaborated in the fourth step?.

Line 112. “The final sample was 224 students…”. Verify number of students (n) between text and table 2.

Line 118. 2.3 Evaluation instrument. The way we see that, this section is sixth step. The information provided on lines 119 to 124 can be presented in the introduction as information about the instrument and its previous development. Was not found how is measured the question 1 to 20, yes/no? Likert scale? See how important this information is when interpreting the data in the results and discussion. It may be interesting that this information is presented.

In 2.5 Data Analysis, on first paragraph it is show a Delphi technique that is usually used to analyze content validity. On second paragraph it is say “Descriptive and inferential statistics…” may became compromised to inferenced the data of this study because the sample was the convenience, had a lot of participant excluded and only was studied a small population.  

RESULTS

Lines 161-162. As mentioned above. The field test was on Spanish students and the expert committee was made up of Brazilian teachers when the target language is Spanish, this could be a limitation and the authors are suggested to reflect on that.

Lines 193-194. The intensity of pain show an average of 4.72 and 4.58 points in the VAS. This intensity of pain seems high, would the authors do some comment in discussion about this? Do the authors think that ICC-correlated data provide relevant information?

DISCUSSION

Lines 221-223. “With respect to… (Table 4), it is suggested to eliminate the concept of reproducibility and revise this statement, because there are some questions with the Kappa coefficient as fair.

Lines 234-236. See comment made on results about this aspect.

Lines 248-251. There are some statements in this paragraph that are difficult to sustain through the data from this study, because your sample is a small convenience population, these data cannot be inferred to the general adolescent population, and further evidence is still needed of validity and reliability to consider this instrument as a gold standard, so it is suggested to review the concepts expressed in this statement.

Line 257. Perhaps the limitations could be revised if some of the suggestions made in this revision are considered.

CONCLUSIONS

Lines 265-266. As the authors planned in the manuscript, the only psychometric test they do is test-retest (unless they consider steps 4-5 as content validity), so when they speak that BackPEI provided valid and reliable score seems an affirmation too strong. They can talk on his process of adaptation to Spanish and its reliability by test-retest.

TABLES

Table 2. The column (n) present 214 students, 108 Female and 106 Male. Review that accord to the text (sample was 224 students).

Table 4. A = Agreement, what type of agreement are you referring to? We do not know the category of each question, for example, see how important it is to know the categorization of questions 19 and 20. Some questions, for example numbers 9 and 10, show a high agreement and a fair Cohen's Kappa coefficient, why did the agreement and the Kappa coefficient present different results? These aspects could be better explained.

Thank you so much.

Reviewer 2 Report

English mus be totally revised, since several mistakes were found

Examples:

p1 line 35 "According this background" the correct is ACCORDING TO

p2 line 45 " which makes its use useful only in those settings." 

p 2 line 47 "then, it could allow a criterion validity study"

Other comments:

p 1 Line 32 "or other IT devices" abbreviation not described

p 1 line 32 "and the poor body use of students (what is this???)

p2 ;ine 61 : whats SL and TL?

Reviewer 3 Report

Manuscript title: Cross-cultural adaptation and reliability of the Back Pain and Body Posture Evaluation Instrument (BackPEI) for Spanish

Comments

I congratulate the authors for conducting such a well-designed research. Please consider the comments below regarding your manuscript reporting.

  1. Abstract: When reporting descriptive statistics, please use a more usual format as mean ± SD. Did you mean ‘4.72 ± 2.33’ and ‘4.58 ± 2.37’)?

  1. Abstract. Consider mentioning the range of kappa values, e.g. from .312 [-,152- ,189] (question #9) to 1.000 (question #13).

  1. Abstract. The conclusion mentions ‘valid’ but no results are shown to support this claim.

  1. Introduction. There are some updated findings regarding low back pain (https://doi.org/10.1007/s00586-014-3571-9) and neck pain (https://doi.org/10.1007/s00586-014-3571-9) in adolescents.

  1. Introduction. The study hypothesis could be rephrased. It suggests that the Spanish version will be compared with the original version in the same sample (or that the ICC values of either version would be compared).

  1. Statistical analysis. What ICC[k,q] model was used? Are the ICC results presented for single or repeated measurements?

  1. Conclusions. The conclusion mentions ‘valid’ but no results are shown to support this claim.

Reviewer 4 Report

Thank you for the opportunity to review this paper. My comments are as follows:

Abstract: Issues with English impact on clarity e.g. first line "has been increasing in the last years" should be "increasing over the last few years" and need to be addressed.

Introduction: Issues with English throughout impact on clarity. Aims of this study need to be more clearly presented.

Materials and Methods: English needs to be addressed as this impacts on the clarity of the description.

Line 73, sentence beginning "However, there is currently..." This step was undertaken in the methodology so this point is probably more appropriate in the Discussion.

Line 76, "comparison of target questionnaire with the original version (from SL to SL)." Should this be (from SL to TL)?

Line 111 point 3) Is the "control name" the participant name? For matching purposes?

Line 119-124 This paragraph would be better at the beginning of the methods to provide context.

Line 128 "Besides, all students voluntarily participated in the study." It is not clear what is meant by this statement. Did students provide consent to participate (in addition to the school and parents)?

Data analysis section is unclear and should be reworked.

Results: Section 3.1 is not clear and should be reworked

Section 3.2 The first sentence repeats what is presented in the methods. Line 195, IC - should this be CI as in confidence interval. And again this section needs to be reworked for clarity.

Discussion: Line 204 "according to this content" It is unclear what this means.  It is unclear what the relevance is of BEHALVES is in relation to a study focussing on translation of the Back PEI. The discussion and conclusion need to be reworked as there is a lack of clarity regarding support for the study aims.

Tables could be clear. Table 1 is difficult to follow and formatting in Table 4 could be improved.